# Shape your Space: A Gaussian Mixture Regularization Approach to Deterministic Autoencoders

**Amrutha Saseendran**[1], **Kathrin Skubch**[1], **Stefan Falkner**[1] **and Margret Keuper**[2]
[1]Bosch Center for Artificial Intelligence
[2]University of Siegen, Max Planck Institute for Informatics, Saarland Informatics Campus
`Amrutha.Saseendran@de.bosch.com`

## Abstract

Variational Autoencoders (VAEs) are powerful probabilistic models to learn representations of complex data distributions. One important limitation of VAEs is the strong prior assumption that latent representations learned by the model follow a simple uni-modal Gaussian distribution. Further, the variational training procedure poses considerable practical challenges. Recently proposed regularized autoencoders offer a deterministic autoencoding framework, that simplifies the original VAE objective and is significantly easier to train. Since these models only provide weak control over the learned latent distribution, they require an ex-post density estimation step to generate samples comparable to those of VAEs. In this paper, we propose a simple and end-to-end trainable deterministic autoencoding framework, that efficiently shapes the latent space of the model during training and utilizes the capacity of expressive multi-modal latent distributions. The proposed training procedure provides direct evidence if the latent distribution adequately captures complex aspects of the encoded data. We show in experiments the expressiveness and sample quality of our model in various challenging continuous and discrete domains. An implementation is available at `https://github.com/boschresearch/GMM_DAE`.

## 1 Introduction

Variational autoencoders (VAEs) constitute one of the popular generative learning frameworks widely used for applications such as image understanding and generation, sentence modeling, and optimizing discrete data and graph-based structures [7, 23, 34, 40, 48]. The VAE framework elegantly combines autoencoders with variational inference [24]. The encoder of the model maps the input data into a lower-dimensional latent space according to a given inference model. The decoder provides a mapping from the latent space back to the original input space. Both are jointly optimized by maximizing a lower bound on the model evidence, regularizing the latent space towards a fixed prior distribution, usually a uni-modal Gaussian. By sampling from the latent space prior, we can utilize the decoder network to efficiently generate new samples from the training distribution. Due to the variational formulation, optimizing the VAE training objective poses significant practical challenges. Further, the over simplistic prior assumption often leads to an unsatisfying trade-off between the quality of reconstructed samples and the prior regularization [2]. Recent work has shown that choosing more flexible priors helps to improve the generative performance of VAEs [44].

Since the initial introduction of VAEs, various novel training objectives have been proposed. One line of work focuses on different regularization techniques derived from alternative probabilistic metrics to shape the latent space of the model during training, e.g. using the Wasserstein distance [43]. In contrast to the KL-divergence, the Wasserstein distance measure induces a metric on probability

distributions. Practically, this facilitates smoother convergence even for initially non-overlapping distributions. Further, it overcomes the *over-regularization effect* in VAEs. To be precise, it prevents the undesired behaviour of multiple data points being mapped to the same latent representation by the encoder. Since closed-form solutions for metrics like the Wasserstein distance can only be derived for very few prior distributions, these approaches rely on numerical approximations during training.

Recent work by Ghosh et al. [12] reinterprets deterministic autoencoders as variational models, even when trained with a deterministic loss. During training, this approach maximizes the negative log-marginal likelihood of the latent samples under a Gaussian normal distribution as a regularization in addition to minimizing the reconstruction loss. Experimental results show that this regularization alone does not suffice to generate high quality samples using the Gaussian prior. To overcome this, Gosh et al. propose to use a multi-modal Gaussian mixture model (GMM) to fit arbitrary, learned latent spaces. While this approach leads to good sampling efficiency and generalization if the post-hoc fit is reasonable, sampling quality can suffer significantly if the learned latent space can not be modeled well by a GMM.

In this work, we propose a deterministic training scheme for autoencoders that is applicable to expressive priors and overcomes the necessity of a post-hoc density estimation step for deterministic training. To be precise, we derive a deterministic regularization loss from the distance metric used in the non-parametric Kolmogorov-Smirnov (KS) test for equality of probability distributions. The resulting training objective can be derived in closed form for a class of expressive multi-modal prior distributions and provides a strong signal to efficiently shape the latent space of the model during training. We chose our experiments to evaluate the proposed approach in terms of sampling quality and expressiveness. In the first line of experiments, we compare the quality of newly generated and reconstructed samples from our model with those from a variety of other VAE variants. In the second line, we investigate our method's capability to model discrete and complex structured inputs such as arithmetic expressions and molecules. In these domains, VAEs have recently been proposed as a tool for dimensionality reduction in optimization. Applying our regularization scheme effectively utilizes multi-modal prior distributions in this context and significantly improves optimization performance.

## 2 Related Work

Since the introduction of VAEs, many follow up works tried to overcome the practical and theoretical limitations of the framework, e.g. [2, 43, 44], and make them applicable to specific applications such as clustering [6, 39] or anomaly detection [49]. We first review some seminal examples of VAE models with different priors and probability metrics for latent regularization. Since our proposed regularization term structures the latent space to a Gaussian mixture model, we also compare it to prior work on deep clustering. Lastly, we discuss VAEs in the context of black-box optimization approaches such as Bayesian Optimization (BO).

**VAEs** In the standard VAE framework, the prior distribution is commonly assumed to be a Gaussian normal distribution. This might lead to simplified representations learned by the model which is unable to represent the rich semantics in the data distribution. Several methods were proposed to introduce more flexible and expressive priors to the VAE formulation. Casale et al. [3] employ Gaussian process priors to account for correlations between the data samples. In [15], a Bayesian non-parametric prior is used with a hierarchical non-parametric variational autoencoder for video representation learning. Chen et al. [4] use an auto-regressive prior to achieve improved generative performance on image datasets. Berger et al. [2] propose to replace the standard spherical Gaussian prior with a more general version with an arbitrary covariance matrix and learn the correlations by optimizing the evidence lower bound of the model. Although the proposed methods offer competitive performance, they often employ complex architectures [4] to achieve desired performance.

In another line of work, multi-modal priors were utilized in VAE models. Zong et al. [49] propose to use a GMM prior in autoendocers for unsupervised anomaly detection by training an additional network estimating the parameters of the GMM. Lee et al. [27] address unsupervised meta-learning using a GMM prior in VAEs to shape the latent space by employing an extension of the evidence lower bound to complex variational inference schemes. Tomczak et al. [44] propose to replace the GMM prior by a coupling of the posterior and prior of the model. Adversarial autoencoders [31] improve the generative performance of VAEs by incorporating adversarial learning into the VAE framework and offer competitive performance in image generation at an increased computational complexity and

decreased training stability. To account for the over regularization effect of the KL divergence term in the standard VAE framework, [43] minimize the Wasserstein distance between the representations learned by the model and the target prior. Recently introduced regularized autoencoders [12](RAEs), question the variational framework adopted by the VAEs and propose a deterministic approach to achieve comparable or better image generation performance. The authors use the negative log likelihood for regularization, but require a post-hoc step to derive a strong sampling procedure from the model. The state of the art VAE model for high fidelity image generation, VQ-VAE [37, 41], can be also considered as a deterministic autoencoder. Similar to RAEs, training VQ-VAE involves two stages of training relying on complex discrete autoregressive density estimators. Moreover, the training loss of VQ-VAE is non-differentiable due to the quantization of the latent vector.

Our approach elegantly combines the idea of new training objectives with the extension to multi-modal priors without increasing training complexity or compromising sampling quality. We derive a strong training signal which can be derived in closed form for multimodal priors. This ensures stable training and reliable regularization of the latent space, improving sampling quality.

**Deep Clustering** Deep Clustering approaches benefit from well structured latent spaces. Thus, several methods employ Gaussian mixture VAEs for data encoding [6, 39] or establish a GMM-like latent space structure through $k$-means models in the latent space. For example, Xie et al. [46] train an autoencoder and apply a KL-divergence loss for better $k$-means clustering while Ghasedi et al. [8] combine the autoencoder reconstruction loss with the relative cluster entropy. Similar approaches have been proposed in the literature [8, 16, 18, 22, 42, 47]. Caron et al. [33] iteratively group points using $k$-means during optimization. In the context of clustering, generative adversarial networks have been considered in [11, 35]. While we are not considering the clustering task in this paper, we hypothesise that the proposed regularization can be beneficial in this context since it implicitly optimizes for mode assignments.

**Structural VAEs and optimization** High-dimensional optimization problems in structured discrete input domains are ubiquitous. VAEs have been used in this context to learn low dimensional, continuous representations of high dimensional, structured data like molecules or arithmetic expressions. Recent work proposes to use such representations to perform efficient optimization by running BO in the latent space of VAEs [25, 30]. In this setting, prior knowledge of the structure of the latent space is crucial to allow for an efficient exploration and generation of valid samples. Yet, as discussed above, VAEs can suffer from simplistic prior assumptions. Thus, sampling from the latent space of such models can result in invalid samples, reducing the sampling efficiency of BO [17]. Kusner et al. [25] overcome this issue if data follows a specific grammar. Lu et al. [30] propose a VAE that directly works on parse trees from context-free grammars to represent discrete data. Yet, those only work with unimodal priors which limits the generalization capabilities. Our approach can be readily used to extend these models to better encode structural data and improve BO performance.

## 3 Method

We introduce a novel loss function to regularize the latent representation learned by deterministic autoencoders towards a given prior distribution. The definition of our loss builds on the non-parametric statistical Kolmogorov-Smirnov (KS) test for equality of one-dimensional probability distributions. We propose a multivariate variant of the distance measure used in the KS test, that allows for gradient based optimization and can easily be applied to expressive multi-modal prior distributions. For ease of exposition, we start with introducing our regularization loss for unimodal Gaussian priors in section 3.1 and extend the formulation to expressive multi-modal Gaussian mixture models in Section 3.2. Finally, in Section 3.3, we provide an explicit way to estimate the weighting parameters of our loss.

### 3.1 Uni-Modal latent regularization

The KS test can be used to determine whether a collection of $N$, one-dimensional samples follow a given reference distribution. It compares the cumulative distribution function (CDF) of the reference distribution with the empirical CDF $\bar{F}^{(N)}$ of the samples. It is often applied to the class of one-dimensional Gaussian distributions, which has important analytical properties. For spherical Gaussians, the one-dimensional KS test quantifies a distance between the empirical distribution

function of the data and the cumulative distribution function

$$\Phi(z) = \frac{1}{\sigma\sqrt{2\pi}} \int_{-\infty}^{x} \exp \frac{-(t-\mu)^2}{2\sigma^2}\, dt \tag{1}$$

of the univariate Gaussian $Z \sim \mathcal{N}(\mu, \sigma)$ as $\sup_{z \in \mathbb{R}} |\bar{F}^{(N)}(z) - \Phi(z)|$. Extending this *KS distance* to higher dimension is particularly challenging, since it requires matching *joint* CDFs [10, 14, 38]. Especially in higher dimensions this becomes infeasible [29]. The continuous ranked probability score [13] shares the same theoretical basis as the KS distance. However it tests whether two sets of samples are consistent with each other, i.e., they could originate from the same distribution, and is thus not suitable to regularize a collection of latent samples towards a given prior distribution. Alternative multi-variate normality tests, like the Mardia test [32] and the BEHP test [1] suffer from slow convergence rates.

To derive a regularization loss from the KS distance, we propose to overcome this issue by taking into consideration the *marginal* CDFs and correlations in the prior distribution separately. Given $d$-dimensional latent samples $\mathbf{z}_1, \ldots, \mathbf{z}_N$, the empirical marginal CDF in dimension $j$ is given by

$$\bar{F}_j^{(N)}(z) = \frac{1}{n} \sum_{n=1}^{N} \mathbb{1}_{[\mathbf{z}_n]_j \leq z}. \tag{2}$$

We aim to regularize the latent space of our models by comparing the empirical marginal CDFs with the one-dimensional CDFs of the marginal distributions of the prior. To strengthen the training signal of our regularization scheme and make it suitable for gradient-based optimization, we replace the supremum in the original KS distance by a smoother MSE loss, that compares the distances between those functions at the latent representations. For a uni-modal Gaussian prior with mean $\boldsymbol{\mu}$ and covariance matrix $\boldsymbol{\Sigma}$, this results in

$$\mathcal{L}_{\text{KS}}(\mathbf{z}_{1,\ldots,N}) = \frac{1}{d} \sum_{j=1}^{d} \text{MSE}\left(\bar{F}_j^{(N)}(\mathbf{z}_j), \Phi(\bar{\mathbf{z}}_j)\right), \qquad \bar{\mathbf{z}}_j = \frac{\mathbf{z}_j - \boldsymbol{\mu}_j}{[\boldsymbol{\Sigma}]_{j,j}}. \tag{3}$$

Here, $\bar{F}_j^{(N)}(\mathbf{z}_j)$ denotes the vector with entries $\bar{F}_j^{(N)}([\mathbf{z}_i]_j)$ and $\Phi(\bar{\mathbf{z}}_j)$ is defined accordingly. This loss is minimized, if the empirical marginal CDFs of the latent samples match those of the uni-modal Gaussian prior. Using the above loss alone will not account for correlations between different latent dimensions. In the case of a spherical Gaussian prior with identity covariance matrix for example, samples with perfectly correlated Gaussian components $[\mathbf{z}_i]_j = [\mathbf{z}_{i'}]_j$, will also minimize this objective, see Figure 1. To overcome this problem, we equip our loss with an additional term, that matches covariances between different latent distributions explicitly. Following a similar reasoning to the MSE above, we define an additional loss term,

$$\mathcal{L}_{\text{CV}}(\mathbf{z}_{1,\ldots,N}) = \frac{1}{d^2} \sum_{l,j=1}^{d} \left([\bar{\boldsymbol{\Sigma}}]_{l,j} - [\boldsymbol{\Sigma}]_{l,j}\right)^2, \tag{4}$$

where $\bar{\boldsymbol{\Sigma}}$ is the empirical covariance matrix of the latent representations and $\boldsymbol{\Sigma}$ stands for the prior covariance. Compared to the negative log marginal regularization proposed in [12], our loss will actually enforce the latent representations to be spread across the entire support of the Gaussian prior, instead of being minimal when all latent collapse to the origin.

### 3.2 Multi-Modal latent regularization

One advantage of our approach is the applicability to more expressive, multi-modal prior distributions. While the Gaussian distribution has important analytical properties, it suffers from significant limitations when modelling real data sets. In contrast, a linear combination of Gaussians can give rise to very complex densities while still allowing for closed form computations of important quantities, like CDFs and covariances. A $d$-dimensional $K$-modal Gaussian mixture model is a weighted super-position of $K$ Gaussian distributions in $\mathbb{R}^d$, that are often referred to as the modes of the model. For $k \leq K$, let $\boldsymbol{\mu}_k$ and $\boldsymbol{\Sigma}_k$ be the mean and covariance matrix of the $k$-th mode in the model. Further, let $p_k > 0$ be the weight of the $k$-th mode. Then, the marginal CDFs of a GMM model can be computed

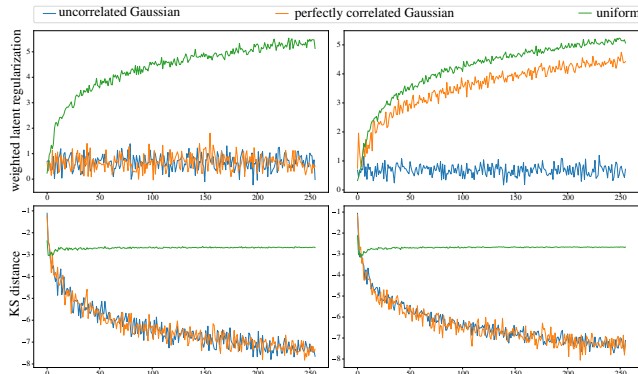

Figure 1: Uni-modal latent regularization in one and two dimensions for varying numbers of samples (x-axis) from different distributions: In two dimensions (right), the simplistic KS distance can not differentiate the target prior (blue) from other probability distributions. By contrast, our proposed regularization scheme successfully matches correlations across different dimensions.

from the CDFs of univariate Gaussians as follows

$$F_{\mathrm{GMM},j}(z) = \sum_{k=1}^{K} p_k \Phi\left(\frac{z - [\boldsymbol{\mu}_k]_j}{[\boldsymbol{\Sigma}]_{j,j}}\right),$$ (5)

i.e. the marginal CDFs in the GMM are weighted sums of CDFs of one-dimensional Gaussians. The covariance matrix of the GMM can be computed as

$$\boldsymbol{\Sigma}_{\mathrm{GMM}} = \sum_{k=1}^{K} p_k \boldsymbol{\Sigma}_k + \sum_{k=1}^{K} p_k \left(\boldsymbol{\mu}_k - \bar{\boldsymbol{\mu}}\right)\left(\boldsymbol{\mu}_k - \bar{\boldsymbol{\mu}}\right)^T, \qquad \bar{\boldsymbol{\mu}} = \frac{1}{k}\sum_{k=1}^{K}\boldsymbol{\mu}_k.$$ (6)

Extending our proposed regularization scheme to multimodal GMMs is straight forward. Our first loss term is defined as

$$\mathcal{L}_{\mathrm{KS},K}(\mathbf{z}_{1,\dots,N}) = \frac{1}{d}\sum_{j=1}^{d}\mathrm{MSE}\left(\bar{F}_j^{(N)}(\mathbf{z}_j), F_{\mathrm{GMM},j}(\mathbf{z}_j)\right).$$ (7)

Similarly, the second loss term is defined to be

$$\mathcal{L}_{\mathrm{CV},K}(\mathbf{z}_{1,\dots,N}) = \frac{1}{d^2}\sum_{l,j=1}^{d}\left([\bar{\boldsymbol{\Sigma}}]_{l,j} - [\boldsymbol{\Sigma}_{\mathrm{GMM}}]_{l,j}\right)^2.$$ (8)

The total loss of the model is a combination of the reconstruction loss and a regularization loss, that enforces the latent representations of the encoded data to match a predefined multi-modal prior distribution. The reconstruction loss $\mathcal{L}_{\mathrm{REC}}(\mathbf{x}'_{1,\dots,N})$ equals the mean squared error between inputs $\mathbf{x}_i$ and their reconstructions $\mathbf{x}'_i$. Given positive weights $\lambda_{\mathrm{KS}}$ and $\lambda_{\mathrm{CV}}$, our final loss is given by

$$\mathcal{L}(\mathbf{x}_{1,\dots,N}) = \lambda_{\mathrm{REC}}\mathcal{L}_{\mathrm{REC}}(\mathbf{x}'_{1,\dots,N}) + \lambda_{\mathrm{KS}}\mathcal{L}_{\mathrm{KS},K}(\mathbf{z}_{1,\dots,N}) + \lambda_{\mathrm{CV}}\mathcal{L}_{\mathrm{CV},K}(\mathbf{z}_{1,\dots,N}).$$ (9)

Formally, the weights $\lambda_{\mathrm{KS}}, \lambda_{\mathrm{CV}}$ and $\lambda_{\mathrm{REC}}$ are hyperparameters of the model. Nevertheless, we propose an explicit way to set $\lambda_{\mathrm{KS}}$ and $\lambda_{\mathrm{CV}}$ and a simple heuristic to estimate $\lambda_{\mathrm{REC}}$ to avoid an extensive optimization of these weights.

### 3.3 Loss weight estimation

Balancing the two regularization losses appropriately poses a key challenge as they potentially vary on very different scales. For example, if modes of the GMM prior are far spread, the covariance $\mathcal{L}_{\mathrm{CV},K}$ loss will dominate the marginal CDF $\mathcal{L}_{\mathrm{KS},K}$ loss by far. Nevertheless, given a target GMM prior, the

dimension of the latent space and the batch size $n$ used during training, there is a concise way to fix those hyperparameters beforehand. To be precise, for $m = 1, \ldots, M$ samples $\mathbf{z}_1^{(m)}, \ldots, \mathbf{z}_N^{(m)}$ from the prior GMM, we propose to set

$$\lambda_{\mathrm{KS}}^{-1} = \frac{1}{M}\mathcal{L}_{\mathrm{KS}}\left(\mathbf{z}_{1,\ldots,N}^{(m)}\right), \qquad\qquad \lambda_{\mathrm{CV}}^{-1} = \frac{1}{M}\mathcal{L}_{\mathrm{CV}}\left(\mathbf{z}_{1,\ldots,N}^{(m)}\right). \qquad (10)$$

Formally, we can not overcome the necessity of tuning the weight of the reconstruction loss, which has significant impact on performance of the model. Nevertheless, a reasonable approximation to it can be obtained by training an autoencoder model and using the inverse of the best obtained loss for $\lambda_{\mathrm{REC}}$. Using this scaling, all loss terms in our regularization loss will ultimately converge to one if the target prior is matched successfully.

## 4    Experiments

With our experiments we strive to investigate the potential of the proposed model when compared to other VAE variants in generating new samples, analyse the effect of the defined prior to effectively cluster the latent space and to shape the latent space efficiently in highly structured domains such as discrete spaces. We provide all the experimental settings and hyperparameters used in the Appendix. All experiments were run on a GPU cluster, with single GPU per individual experiments. Since the cluster is part of a carbon-neutral framework, these experiments did not contribute to climate change.

### 4.1    Image generation

We consider four dataset, MNIST [26], FASHIONMNIST [45], SVHN [36] and CELEBA [28] to evaluate the proposed method in image generation experiments. The qualitative analysis of the generated samples for MNIST, SVHN amd CELEBA images are shown in Figure 2 along with the reconstructed samples and interpolated samples in the latent space of the trained model. In order to assess the quality of the generated images, we evaluate the Fréchet Inception Distance (FID) [19] for each dataset, see Table 1. For baseline comparison, we evaluate the following models: vanilla variational autoencoder (VAE [24]), Gaussian mixture variational autoencoder (GMVAE) [6], Wasserstein autoencoder (WAE) [43] with MMD loss, 2stage VAEs (2s-VAE) [5], constant variance-VAE (CV-VAE) [12] and regularized autoencoders (RAEs) [12]. We consider the following evaluation metrics: 1. Sampling FID (Samp.) - FID score of the generated random samples (evaluated by generating random samples from the prior distribution of the respective models and by fitting a Gaussian distribution to models trained without any prior assumptions), 2. reconstruction FID (Rec.) - measured by computing the FID between the test samples and their corresponding reconstructions by the model and 3. interpolation FID (Inter.) - measured by computing the FID between the interpolated samples in the latent space and test samples. As pointed out by [12], fitting an ex-post density estimator on the learned embedding after training of VAEs further improves the generation quality. Hence, we also report the FID values by fitting a GMM in the learned latent space of the trained model (GMM column in Table 1, not evaluated for 2s-VAE as they perform ex-post density estimation using another VAE).

As shown in Table 1, our method achieves better FIDs (Samp.) on all datasets considered, when compared to all considered baselines sampled by fitting a single Gaussian in the latent space. We also improved the generation quality as argued above by fitting a mixture of Gaussians in the latent space and achieve better FIDs in MNIST, FASHION MNIST and CELEBA images, whereas for SVHN, WAEs achieved the overall best score. It is also important to note that the proposed method performs comparably or even better without employing the ex-post density estimation. The proposed method also achieves better reconstruction quality than the other VAEs except for SVHN images where RAEs performs better. The interpolation FID indicates the overall structure of the learned latent space and the obtained FID values show that the proposed method shapes the latent space better than the other approaches except for the CELEBA images where RAEs performs slightly better than our method. For a fair comparison, we use the same architecture and experimental settings in all the considered baseline evaluations. Please refer to the Appendix for more details on the experimental settings.

### 4.2    Unsupervised image clustering

We evaluate the potential of our method to naturally cluster the data points in the learned latent space in two dataset, MNIST and FASHION-MNIST. The Gaussian mixture model prior with $k$

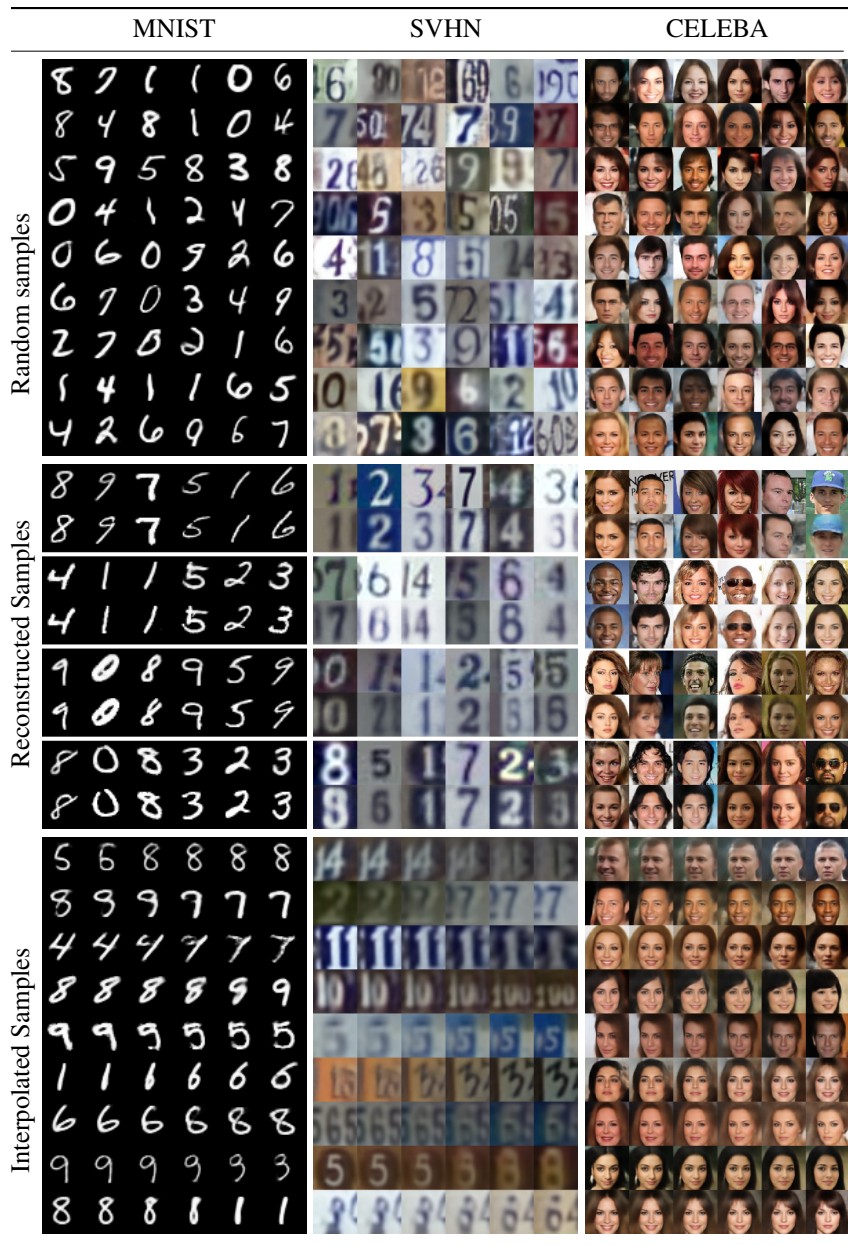

Figure 2: Qualitative analysis on image generation across datasets, MNIST, SVHN and CELEBA. Row 1 shows the randomly generated samples; row 2 shows the reconstructed samples by the decoder on test dataset after training, first row in each sections corresponds to the ground truth and the second one its corresponding reconstruction; row 3 shows randomly interpolated samples in the learned latent space of our model.

components in our method could be considered as $k$ different classes/clusters to which the data points are mapped by the encoder. We train the model with latent space dimension 10 for both dataset and visualize the random samples generated from each Gaussian component of our prior as shown in Figure 3. The figure shows that visually similar images fall into the same cluster. For a quantitative analysis of the clustering performance, we evaluated the unsupervised classification accuracy (similar to [21]) and compare the performance with JointVAE [9] and CascadeVAE [21]. The observed values are reported in the table in Figure 4. We observed a comparable performance to both baselines. We also observed that the distance between the modes in the GMM prior is a deciding factor in better clustering performance. Figure 4 (right) shows the performance comparison of both image generation

Table 1: Quantitative evaluation results across datasets. Samp. refers to the FID of the generated samples from the prior distribution or by fitting a Gaussian to the learned models trained without prior, GMM refers to the FID computed by fitting GMM on the learned model, Rec. refers to the reconstruction FID on test samples and Inter. refers to the Interpolation FID.

| Dataset | MNIST | | | | FASHION MNIST | | | |
|---|---|---|---|---|---|---|---|---|
| | Samp. | GMM | Rec. | Inter. | Samp. | GMM | Rec. | Inter. |
| VAE | 27.27 | 20.52 | 21.59 | 21.05 | 50.50 | 36.22 | 33.33 | 44.12 |
| GMVAE | 21.35 | – | 20.64 | 20.21 | 40.23 | – | 38.79 | 38.54 |
| WAE | 20.20 | 12.90 | 14.07 | 16.19 | 39.66 | 28.01 | 24.84 | 35.01 |
| CV-VAE | 32.12 | 28.62 | 29.61 | 30.76 | 57.57 | 38.28 | 35.10 | 47.73 |
| 2sVAE | 26.99 | – | 23.77 | 22.13 | 46.47 | – | 31.93 | 41.06 |
| RAE | 17.72 | 14.15 | 14.69 | 15.57 | 47.26 | 29.59 | 24.54 | 34.77 |
| Ours | **13.11** | **12.82** | **8.99** | **12.82** | **33.70** | **26.62** | **19.56** | **29.17** |

| Dataset | SVHN | | | | CELEBA | | | |
|---|---|---|---|---|---|---|---|---|
| | Samp. | GMM | Rec. | Inter. | Samp. | GMM | Rec. | Inter. |
| VAE | 61.01 | 58.23 | 59.13 | 50.29 | 68.01 | 61.63 | 52.55 | 58.39 |
| GMVAE | 49.74 | – | 48.65 | 47.15 | 65.35 | – | 64.22 | 64.92 |
| WAE | 58.08 | **34.87** | 29.62 | 27.16 | 58.91 | 49.17 | 41.14 | 47.08 |
| CV-VAE | 51.01 | 54.19 | 48.53 | 47.65 | 57.61 | 52.72 | 45.32 | 50.87 |
| 2sVAE | 45.84 | – | 44.27 | 40.23 | 53.12 | – | 44.78 | 47.64 |
| RAE | 42.35 | 35.12 | **31.04** | 27.30 | 52.33 | 48.23 | 41.61 | **46.58** |
| Ours | **37.42** | 36.46 | 31.27 | **24.87** | **49.79** | **44.79** | **39.48** | 47.13 |

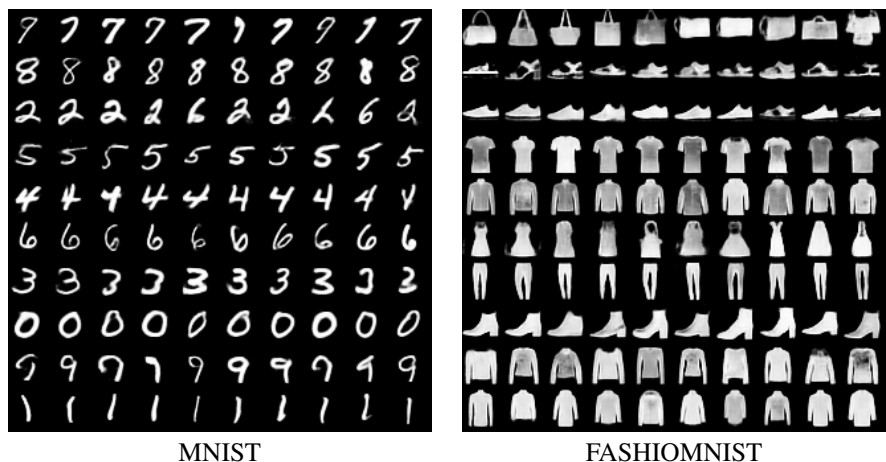

MNIST                                 FASHIOMNIST

Figure 3: Clustering performance on MNIST and FASHION-MNIST images with a 10 component GMM prior. Each row in the figure shows randomly generated images from different Gaussian components of the GMM prior. Similar looking images are mapped into same clusters.

and clustering performance with increasing distance between different modes in the GMM prior. The result shows that with increasing distance, the clustering performance is improved wheres the quality of the generated images gets reduced. Our experimental analysis indicates that natural clustering happens with the multi-modal GMM prior in our method.

## 4.3 Modelling discrete data structures

In this section, we investigate the ability of our model to generate complex discrete data structures such as arithmetic expressions and molecules. The objective of this experiments is to analyze the model performance on shaping the latent space of such structured discrete spaces effectively. The learned latent space of the model is traversed to generate new samples with the desired properties by performing Bayesian Optimization (BO). We perform experiments in two sequence optimization problems similar to [25].

| Method | Acc(↑) | |
|---|---|---|
| | MNIST | FASHION-MNIST |
| JointVAE | 78.33 | 51.51 |
| CascadeVAE | 84.19 | **57.72** |
| Ours | **85.53** | 56.24 |

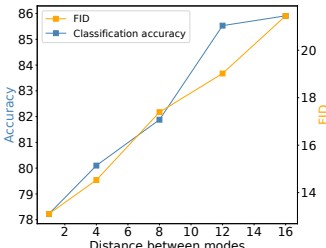

Figure 4: Image clustering: (left) Unsupervised classification results on MNIST and FASHION-MNIST images, (right) Image clustering(Accuracy) and generation performance(FID) on MNIST images with increase in the distance between modes in the GMM prior.

**Arithmetic Expression** Given a dataset of $50,000$ univariate (functions of $x$) arithmetic expressions following a formal grammar [25], the task is to find the expression that best fits a target dataset. This is done by minimizing $\log(1 + \text{MSE})$, where the MSE is computed between values of the generated expression and the target points. For our evaluation, we choose similar target data points as in [25].

**Chemical Design** Given the ZINC250k dataset of drug molechules [20], the objective is to generate new drug like molecules. The drug likeliness of a molecule is quantified by the water-octanol partition coefficient, which is maximized in our line of experiments.

**Results** We extend the architecture and experimental settings of [25] to include our proposed losses during training. For baseline comparison, we consider Grammar VAE (GVAE) [25], Character VAE (CVAE) [17], Grammar constant variance VAE (GCVVAE) [12] and Grammar based RAE (GRAE) [12] frameworks. The three best scores found by our method for arithmetic expressions and the molecule experiments are reported in Table 2. Our model performs comparatively better than the considered baselines and achieves the best first score for both tasks. In addition to the optimization performance, it is also important to consider the validity of the new samples generated by the models. A well-structured latent space should yield valid samples following the defined grammar/rules of the used dataset. Our model achieves better validation and average scores as shown in Table 3 except for GCVVAE which achieves a better average score in the arithmetic expression task. All reported values are evaluated by averaging across 5 BO trials.

Table 2: Best scores found by each method for arithmetic expression and molecule experiments. Baseline values reported from [12].

| Method | Expressions | | | Molecules | | |
|---|---|---|---|---|---|---|
| | 1st(↓) | 2nd(↓) | 3rd(↓) | 1st(↑) | 2nd(↑) | 3rd(↑) |
| GVAE | 0.10 | 0.46 | 0.52 | 3.13 | 3.10 | 2.37 |
| CVAE | 0.45 | 0.48 | 0.61 | 2.75 | 0.82 | 0.63 |
| GCVVAE | 0.39 | 0.40 | 0.43 | 3.22 | 2.83 | 2.63 |
| GRAE | 0.39 | **0.39** | 0.43 | 3.74 | 3.52 | **3.14** |
| Ours | **0.03** | 0.40 | **0.41** | **4.15** | **3.84** | 3.12 |

Table 3: Fraction of valid samples and their corresponding average scores for arithmetic expression and molecule experiments for each method. Baseline values reported from [12].

| Method | Expressions | | Molecules | |
|---|---|---|---|---|
| | Frac. valid (↑) | Avg. score (↓) | Frac. valid (↑) | Avg. score (↑) |
| GVAE | 0.99 ± 0.01 | 3.26 ± 0.20 | 0.28 ± 0.04 | -7.89 ± 1.90 |
| CVAE | 0.82 ± 0.07 | 4.74 ± 0.25 | 0.16 ± 0.04 | -25.64 ± 6.35 |
| GCVVAE | 0.99 ± 0.01 | **2.85 ± 0.08** | 0.76 ± 0.06 | -6.40 ± 0.80 |
| GRAE | **1.00 ± 0.00** | 3.22 ± 0.03 | 0.72 ± 0.09 | -5.62 ± 0.71 |
| Ours | **1.00 ± 0.00** | 3.32 ± 0.04 | **0.72 ± 0.03** | **-5.08 ± 1.30** |

## 4.4 Ablation study and hyperparameter sensitivity analysis

We perform an ablation study on the regularization loss terms in the proposed model. When the model is trained without the KS distance loss for MNIST images, we observed an FID of $49.82$ and when trained without the covariance matching loss, we observed an FID of $38.45$. These values are significantly worse than the FID that we achieve when training with the weighted combination of both regularization losses i.e $13.11$. These empirical evaluations show that the combination of the two regularization terms facilitates a better prior-posterior match and hence better image generation. Please refer to the Appendix for qualitative evaluation of the ablation study on MNIST images.

From a conceptual point of view, the most important hyperparameters of our model are (a) the weights of the different terms in the training objective and (b) the number of components in the prior. In section 3.4, we propose an explicit way how to fix the weights in the loss function. We investigate the sensitivity of our model performance to the number of components in the GMM prior. We trained our model on the MNIST dataset using a GMM prior with $1, 5, 10, 15,$ $20$ and $25$ modes respectively. The observed FID scores for the respective number of components are shown in Figure 5. The result show that with increasing number of components in the chosen prior, the performance of our model improves significantly. As a consequence, choosing a large number of components can be beneficial for practical considerations.

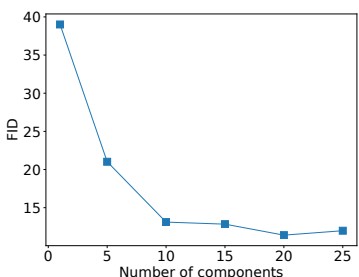

Figure 5: Hyperparameter sensitivity analysis - FID of the MNIST generated samples when model is trained with different number of components in the GMM prior.

## 5 Limitations and Future Work

One limitation of our work is the necessity to chose the prior distribution in advance. We showed that fixing a suitable number of modes for the GMM is important to provide better sampling quality. Also, by considering marginal CDFs, we simplified the original distance metric from the KS test. While reducing computational complexity during training, this comes at the cost of an additional loss term. Further, our proposed addition of the KS distance is not suitable for matching higher order moments of the latent representations to the target prior, which at least from a conceptual point of view can lead to a mismatch to the prior. Further, our loss only facilitates matching empirical marginal CDFs of latent representations to the marginal CDFs of the prior evaluated at latent vectors. As a consequence, our regularization loss might be a less stable training signal for small batch sizes in high dimensions.

Using the Frobenius norm in the covariance matching loss reflects our assumption that the latent dimensions should all be independent from each other and should all simultaneously match the prior's values. The choice for the MSE for covariance matching loss is purely based on its prevalence in the literature. Additionally, this makes all three loss terms (reconstruction loss, KS distance loss and the covariance matching loss) behave similarly, as they are all squares. While we did not investigate any other metrics for matrix comparison in this scenario, exploring other options for the covariance matching is an interesting area for future studies. We have not considered the case where there exists class imbalance in the dataset. We would expect the model to separate the classes if the imbalance is weak and the classes are sufficiently different such that the reconstruction loss outweighs the regularization penalty for the mismatch. Extending our prior to accommodate for this by introducing a weighted GMM prior is also a very interesting direction for future work.

## 6 Conclusion

Recent studies have illustrated the effectiveness of flexible priors in VAEs to learn more meaningful latent representations. Following recent work that highlights the potential of deterministic alternatives to the variational formulation in VAEs, we propose a simple deterministic autoencoding framework with more powerful regularizers to accommodate for expressive multi-modal priors. In particular, we derive a novel deterministic regularization scheme from a strong metric on probability distributions. The proposed approach can be readily applied to effectively shape the latent space of existing autoencoding frameworks towards multi-modal Gaussian priors. Our experimental evaluations show that the proposed training objective yields comparable sampling quality to those of variational autoencoders and achieves better performance in modelling complex discrete data structures.

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
