# Shape your Space: A Gaussian Mixture Regularization Approach to Deterministic Autoencoders

**Amrutha Saseendran**[1]**, Kathrin Skubch**[1]**, Stefan Falkner**[1] **and Margret Keuper**[2]
[1]Bosch Center for Artificial Intelligence
[2]University of Siegen, Max Planck Institute for Informatics, Saarland Informatics Campus
Amrutha.Saseendran@de.bosch.com

## A    Appendix

In this document, we provide additional details and results to the main paper. The document is structured as follows:

A.1 Loss Analysis - Analysis of the unimodal and multimodal latent regularization loss across different distributions and an ablation study on the proposed loss function.

A.2 Image Generation - In this section, we compare VQVAE model with our method, provide detailed descriptions of the dataset, network architecture, and implementation details of the image generation experiments in the main paper.

A.3 Modelling Discrete Structures - In this section, we describe the experimental and implementation details of the discrete data structure experiments in the main paper.

A.4 Mode Analysis - Quantitative analysis of the clustering performance in MNIST and FASH-ION MNIST images.

A.5 Additional Qualitative Analysis - More examples of the randomly generated samples of MNIST, FASHION MNIST, SVHN and CELEBA images.

### A.1    Loss Analysis

**Unimodal Regularization loss**    In this section, we analyze the unimodal version of the proposed regularization loss across different distributions. For a fixed, target prior we investigate the behaviour of our loss on samples from varying distributions. Throughout the unimodal analysis, we choose the target prior to be a standard normal distribution. In our experiments, we evaluate our loss on samples from unimodal Gaussian distributions with (1) standard deviation equal to the prior, but different means and (2) mean equal to the prior, but varying standard deviation. The observed values for the proposed weighted regularization loss are plotted in Figure 1 for dimensions 1 and 2. It can be observed that the loss function increases with increasing distance between the means and standard deviations of the sampling distribution and the target prior.

**Multimodal Regularization loss**    Next, we extend the analysis to the multimodal regularization loss across different distributions. Throughout the multimodal analysis, we fix the target prior to be a Gaussian mixture model with two equally weighted spherical components centered at one hot encoding vectors of the respective dimensions. Similar as above, we consider two sets of experiments for the evaluation: (1) Samples from a GMM with two spherical components centered at different means and (2) Samples from a GMM with two components centered at the means of the prior components, but different covariance. In both cases, we vary the means and covariances by adding a multiplicative factor $\alpha$ and $\beta$ to the means and covariance matrices of the prior respectively. The observed values for the proposed weighted regularization loss are plotted in Figure 2 for dimensions

35th Conference on Neural Information Processing Systems (NeurIPS 2021).

2 and 3. It can be observed that the value of the loss function increases with the increasing factors $\alpha$ and $\beta$.

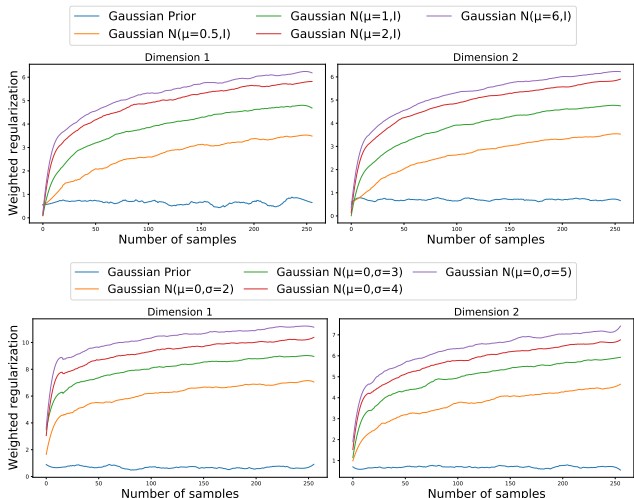

Figure 1: Loss analysis - Unimodal latent regularization in one and two dimensions for varying numbers of samples from different Gaussian distributions. With increase in mean and standard deviation the loss function increases with respect to the target prior (blue).

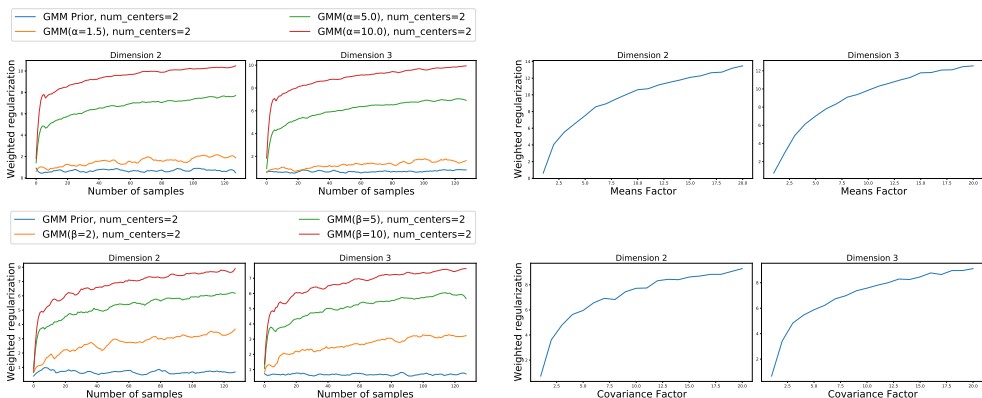

Figure 2: Loss analysis - left: Multimodal latent regularization in two and three dimensions for varying numbers of samples from different Gaussian mixture distributions. Right: Multimodal latent regularization in two and three dimensions for different mean($\alpha$) and covariance factor($\beta$). With increase in mean and covariance of the samples, the loss function increases with respect to the target GMM prior (blue).

**Latent space analysis**   We perform ablation study on the two loss terms in the proposed regularzation loss. For simplicity, we consider a subset of MNIST images with two digits $1$ and $8$ as training dataset for this line of experiments. For ease of visualization, the prior is chosen to be a mixture of two Gaussian with means $(3,3)$ and $(-3,-3)$ and identity covariance matrices. We train a deterministic autoencoder with the two individual loss terms (mean squared covariance distances and simplified Kolmogorov-Smirnov distance) and the proposed weighted combination of both. In Figure 3, we show the latent representations of the training data using these three regularizers after $1$ and $30$ epochs. It can be seen that a combination of the proposed two loss terms is essential for effectively regularizing the latent representations to match the target prior.

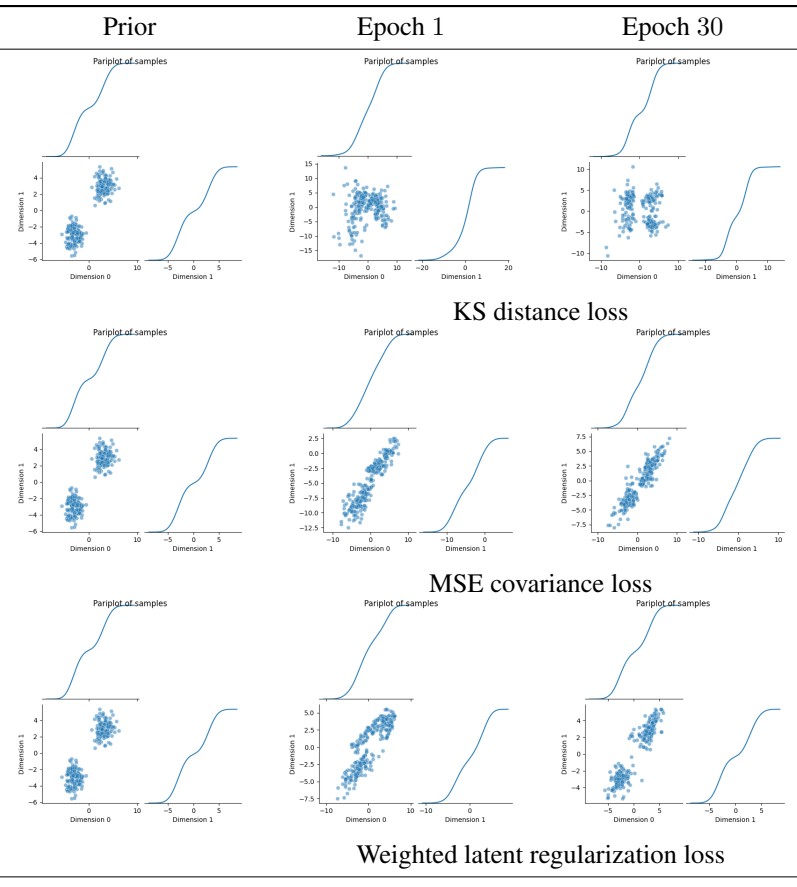

Figure 3: Ablation study on loss functions - 2D pair plot visualization of the target prior and posterior (test images) of the proposed model trained on a subset of MNIST images with different terms of the loss functions.

## A.2 Image Generation

This section summarizes the VQVAE comparison to our model and the implementation details of the image generation experiments in the main paper.

### A.2.1 Comparison to VQVAE

The VQVAE [8, 9] models can be also be considered as a deterministic autoencoder and they focuses on high fidelity image generation. Training VQ-VAE involves two stages of training relying on complex discrete autoregressive density estimators. The training also involves tuning two important hyperparameters, size of the discrete latent space/number of embeddings (K) and the dimensionality of each latent embedding vector (D). We applied VQVAE [8] to MNIST data and observed the following results: for $K = 32$ and $D = 128$, we observed a sampling FID of $18.51$, reconstruction FID of $17.62$ and interpolation FID of $16.85$. However, VQVAE does not address the question of how to structure the continuous latent space before quantization. In contrast, the proposed approach addresses specifically this question of how to efficiently structure the latent space. In that sense, the two approaches are rather complementary than direct competitors. A combination of both for high quality image generation from pre-structured latent spaces prior to quantization would be an interesting topic for future research.

### A.2.2 Dataset

We performed empirical evaluations for image generation across four dataset, MNIST [5], FASHION-MNIST [10], SVHN [7] and CELEBA [6].

**MNIST** The dataset (licensed under GNU General Public License v3.0) includes gray-scale images of hand-written digits from 0 to 9. The original MNIST images of size $28 \times 28$ were padded to size $32 \times 32$ for all our experiments. The training dataset includes 50000 images, validation and test dataset includes 10000 images each.

**FASHION-MNIST** The dataset (licensed under MIT License) includes gray scale images of fashion and clothing items of 10 classes. The original images of size $28 \times 28$ were padded to size $32 \times 32$ for all our experiments. The training dataset includes 50000 images, validation and test dataset includes 10000 images each.

**SVHN** The dataset (licensed under GNU General Public License v3.0) includes house numbers cropped from street view images (RGB) and is available in two formats, cropped versions of size $32 \times 32$ and original images of varying resolution. For our experiments, we considered the cropped versions. The training dataset includes 73257 images, validation dataset of 10000 images and test dataset of 16032 images.

**CELEBA** The dataset (licensed under Custom(non-commercial research)) includes over 200k celebrity face images (RGB). The training dataset includes 162721 images, validation dataset of 19866 and test dataset 19961 images. The images were center cropped to size $140 \times 140$ and then resized to $64 \times 64$ before training.

### A.2.3 Network architecture: Training details and hyper-parameters

The architectural details of the encoder and decoder used are shown in Table 1. For fair comparison we used the same architecture for all the baseline methods. Filter size of $4$ is used for all layers in the network, with padding size, $1$ and stride $2$. Please refer to the code appendix for the implementation of the proposed model. For regularized autoencoders (RAE) and other VAEs in the baseline comparison, we used the official GitHub repository to evaluate the results[1]. We used the Pytorch implementation in the Github repository[2] for GMVAE experiments.

Table 1: Encoder and Decoder network architecture - Image generation. Conv2D stands for the convolution layer, BN corresponds to batch normalization, Conv2DT refers to the transposed convolution layer and FC stands for fully connected layer.

| Dataset | Encoder | | Decoder | |
|---|---|---|---|---|
| | Layer | Output | Layer | Output |
| MNIST/ | Input | $1 \times 32 \times 32$ | Input | $10 \times 1$ |
| FASHION MNIST | Conv2D, BN, ReLU | $128 \times 16 \times 16$ | FC, Reshape | $1024 \times 2 \times 2$ |
| | Conv2D, BN, ReLU | $256 \times 8 \times 8$ | Conv2DT, BN, ReLU | $512 \times 4 \times 4$ |
| | Conv2D, BN, ReLU | $512 \times 4 \times 4$ | Conv2DT, BN, ReLU | $256 \times 8 \times 8$ |
| | Conv2D, BN, ReLU | $1024 \times 2 \times 2$ | Conv2DT, BN, ReLU | $128 \times 16 \times 16$ |
| | Flatten, FC | $10 \times 1$ | Conv2DT, BN, ReLU | $1 \times 32 \times 32$ |
| SVHN | Input | $3 \times 32 \times 32$ | Input | $100 \times 1$ |
| | Conv2D, BN, ReLU | $128 \times 16 \times 16$ | FC, Reshape | $1024 \times 2 \times 2$ |
| | Conv2D, BN, ReLU | $256 \times 8 \times 8$ | Conv2DT, BN, ReLU | $512 \times 4 \times 4$ |
| | Conv2D, BN, ReLU | $512 \times 4 \times 4$ | Conv2DT, BN, ReLU | $256 \times 8 \times 8$ |
| | Conv2D, BN, ReLU | $1024 \times 2 \times 2$ | Conv2DT, BN, ReLU | $128 \times 16 \times 16$ |
| | Flatten, FC | $100 \times 1$ | Conv2DT, BN, ReLU | $3 \times 32 \times 32$ |
| CELEBA | Input | $3 \times 64 \times 64$ | Input | $64 \times 1$ |
| | Conv2D, BN, ReLU | $128 \times 32 \times 32$ | FC, Reshape | $1024 \times 4 \times 4$ |
| | Conv2D, BN, ReLU | $256 \times 16 \times 16$ | Conv2DT, BN, ReLU | $512 \times 8 \times 8$ |
| | Conv2D, BN, ReLU | $512 \times 8 \times 8$ | Conv2DT, BN, ReLU | $256 \times 16 \times 16$ |
| | Conv2D, BN, ReLU | $1024 \times 4 \times 4$ | Conv2DT, BN, ReLU | $128 \times 32 \times 32$ |
| | Flatten, FC | $64 \times 1$ | Conv2DT, BN, ReLU | $3 \times 64 \times 64$ |

---

[1]https://github.com/ParthaEth/Regularized_autoencoders-RAE-
[2]https://github.com/jariasf/GMVAE

We train our model with ADAM optimizer [3], using a batch size of 100, number of epochs 100, momentum $(\beta_1, \beta_2) = (0.5, 0.999)$ with starting learning rate of $0.002$ which exponentially decays when the validation loss plateaus. The latent space dimension of MNIST and FASHION-MNIST is 10, 100 for SVHN and 64 for CELEBA images. The image reconstruction loss coefficient value of 0.005 is used for all experiments. The other two loss coefficient values can be calculated as mentioned in section 3 of the main paper. For prior definition, we define the means of each Gaussian component as one hot encoding vector with a standard deviation of 1. A mixture of 10 components with equally weighted mixing co-efficients were used for MNIST, FASHION MNIST and SVHN, 20 for CELEBA images. For evaluation metrics, Fréchet Inception Distance (FID) [2] is calculated for 10000 images and averaged across 5 different runs. The FIDs observed by sampling from the prior along with error bars (for different runs) are as follows, MNIST: $13.11 \pm 0.9$, FASHION-MNIST: $33.70 \pm 0.8$, SVHN: $37.42 \pm 1.1$ and CELEBA: $49.79 \pm 1.2$. And the FIDs that we observe after fitting a GMM to the latent space of our model are as follows (for different runs), MNIST: $12.82 \pm 0.6$, FASHION-MNIST: $26.62 \pm 0.8$, SVHN: $36.46 \pm 0.9$ and CELEBA: $44.79 \pm 1.0$. All our experiments were conducted on a single GTX1080 GPU with 12/16 GB RAM memory[3].

## A.3    Modelling Discrete Data Structures

In this section we discuss the experimental setup and implementation details of the structured data experiments in the main paper. We consider two optimization problems similar to [4], 1. searching the latent space for arithmetic expression that best fits a target dataset and 2. searching for the best drug like molecule. We follow the same experimental setup as in [4], which we summarize in the following for clarity.

**Arithmetic expression fitting task** The model is trained with a dataset of $100,000$ randomly generated univariate arithmetic expressions (functions of $x$) following a defined grammar [4]. The objective of this experiment is to search in the latent space of the trained model to find an expression that best matches a fixed target dataset. The target dataset is defined by selecting 1000 input values of x, that are linearly-spaced between $-10$ and 10. The corresponding x values are given to the true function $1/3 + x + \sin(x * x)$ to generate target observations. The target variable/score to optimize is defined as the $\log(1 + \text{MSE})$ between the predictions made by an expression and the true data. Bayesian optimization (BO) is utilized to traverse through the latent space of arithmetic expressions to search for an equation that best matches this target true function. The best three expressions found by our method along with their corresponding score is reported in Table 3.

**Molecule discovery** The model is trained with a dataset of $250,000$ SMILES strings ZINC250K [1] following the context free grammar as defined in [4]. The latent space of the trained model is then traversed to find the molecule with the best drug likeliness score. The drug likeliness score is quantified by the design metric water octanol partition co-efficient (logP) of the molecules. The best three molecules generated by our model along with their target score is given in Table 4.

**Implementation** We extend the official Tensorflow implementation of GRAMMAR-VAE [4][4] with our novel regularizer to evaluate the results. The image reconstruction loss coefficient used is 0.005 for both the experiments. The other two loss coefficient values can be calculated as mentioned in section 3 of the main paper. We used the same network architecture and other hyper-parameters similar to the original implementation.

**Predictive performance of the latent representation** Similar to [4] we also evaluate the predictive performance of the the latent representations of the proposed model. The sparse Gaussian process model used in the BO is used to evaluate the predictive performance on a left out 10% of data (test). The input to the sparse GP model is the test data (formed by the latent representation of the available sequences) and the output is the prediction of the associated properties/scores of each tasks. The test log likelihood and the average RMSE values obtained for our model is compared to GVAE [4] and CVAE [1] in Table 2. Our model yields better predictive performance on both tasks which shows that the proposed model learned better latent features for better predictions compared to the other two baseline models.

---

[3]GPU cluster part of carbon neutral framework

[4]`https://github.com/mkusner/grammarVAE`

Table 2: Predictive performances of sparse Gaussian processes on different VAEs. Baseline values are taken from [4].

| Objective | Method | Expressions | Molecules |
|-----------|--------|-------------|-----------|
| LL | GVAE | $-1.320 \pm 0.001$ | $-1.739 \pm 0.004$ |
| | CVAE | $-1.397 \pm 0.003$ | $-1.812 \pm 0.004$ |
| | Ours | **-1.309 ± 0.001** | **-1.689 ± 0.003** |
| RMSE | GVAE | $0.884 \pm 0.002$ | $1.404 \pm 0.006$ |
| | CVAE | $0.975 \pm 0.004$ | $1.504 \pm 0.006$ |
| | Ours | **0.877 ± 0.001** | **1.400 ± 0.002** |

Table 3: The generated expressions corresponding to the observed best three scores.

| Number | Expression | Score($\downarrow$) |
|--------|------------|---------|
| 1 | $x * 1 + \sin(3) + \sin(x * x)$ | 0.03 |
| 2 | $x * 1 + \sin(1) + \sin(2 * 3)$ | 0.40 |
| 3 | $x + 1 + \sin(3) + \sin(3 + 2)$ | 0.41 |

Table 4: The generated molecules corresponding to the observed best three scores.

| Number | SMILE | Score($\uparrow$) |
|--------|-------|---------|
| 1 | C(CCC)CCCCCCCC | 4.15 |
| 2 | CCCCCCCCCCC | 3.84 |
| 3 | CCCCCc1cccc(c1) | 3.12 |

## A.4 Mode Analysis

Although clustering is not the goal of this paper, we investigate the performance of the model to generate samples from similar classes within each component of the GMM prior. In addition to the qualitative and quantitative analysis of clustering performance of the proposed method in the main paper, we also report the quantitative unsupervised clustering performance in terms of two other metrics, 1. Normalized Mutual Information (NMI) and 2. mean Average Precision (mAP). NMI measures the mutual information between the cluster assignments and the ground truth labels and is normalized by the average of the entropy of both target and observed labels. The calculated NMI and mAP values for the MNIST and FASHION MNIST test images are reported in the Table 5. The images similar in visual appearance are grouped in to same components in the prior and the proposed model achieves reasonable natural clustering of the object classes in both MNIST and FASHION MNIST images.

Table 5: Unsupervised clustering performance on MNIST and FASHION MNIST images.

| Dataset | NMI | mAP |
|---------|-----|-----|
| MNIST | 0.72 | 0.75 |
| FASHION MNIST | 0.60 | 0.61 |

## A.5 Additional Qualitative Analysis

The qualitative analysis of the generated samples for FASHION MNIST images are shown in Figure 4 along with the reconstructed samples and interpolated samples in the latent space of the trained model. More examples of randomly generated MNIST, FASHION MNIST, SVHN and CELEBA images are given in Figure 5 and Figure 6 respectively.

| Random samples | Reconstructed samples | Interpolated samples |

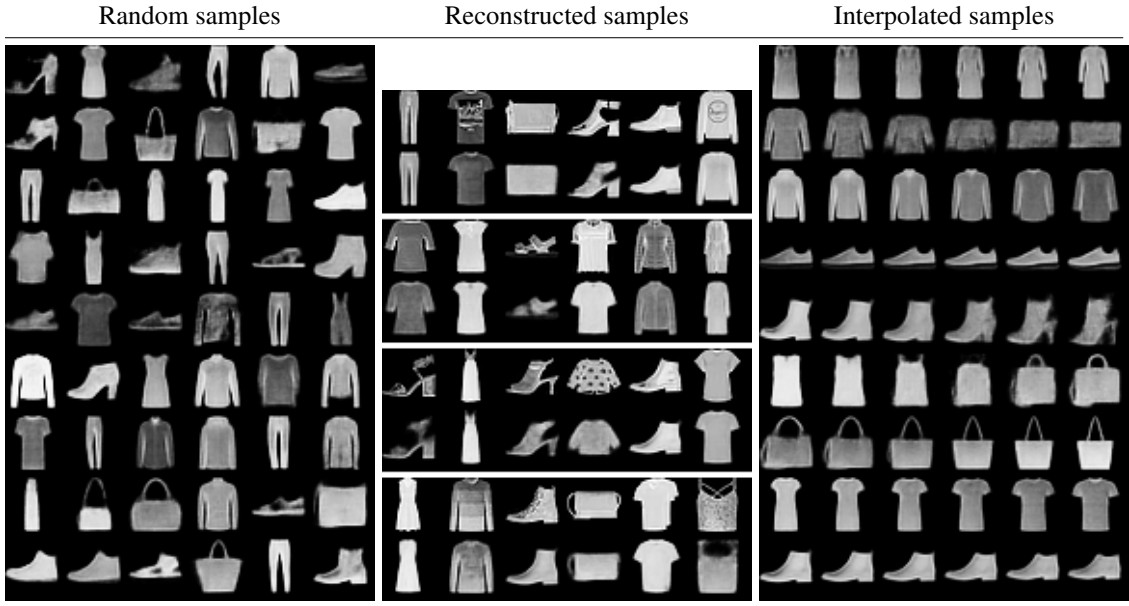

Figure 4: Qualitative analysis on image generation on FASHION-MNIST images. Column 1 shows the randomly generated samples; Column 2 shows the reconstructed samples by the decoder on test dataset after training, first row in each sections corresponds to the ground truth and the second one its corresponding reconstruction; Column 3 shows randomly interpolated samples in the learned latent space of our model.

| MNIST | FASHION MNIST |
|:---:|:---:|

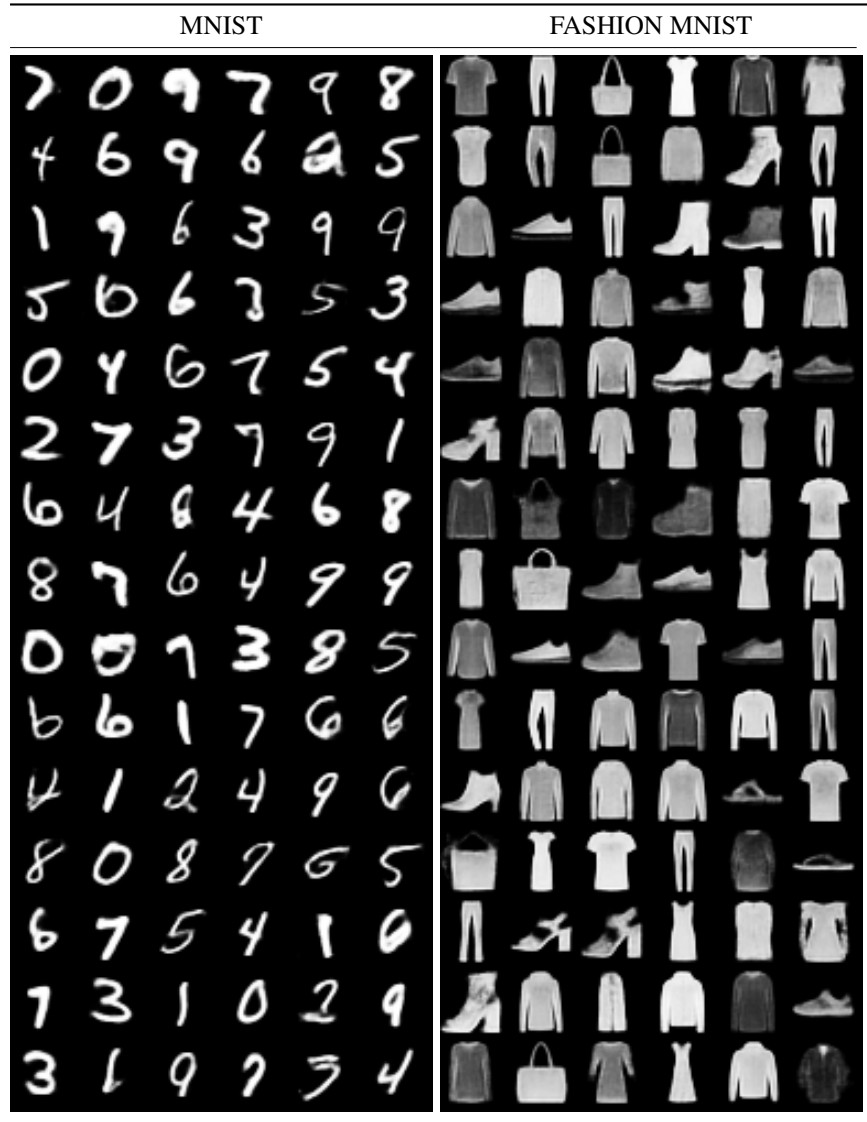

Figure 5: Additional qualitative analysis on image generation - randomly generated MNIST and FASHION MNIST smaples.

| SVHN | CELEBA |
|:---:|:---:|

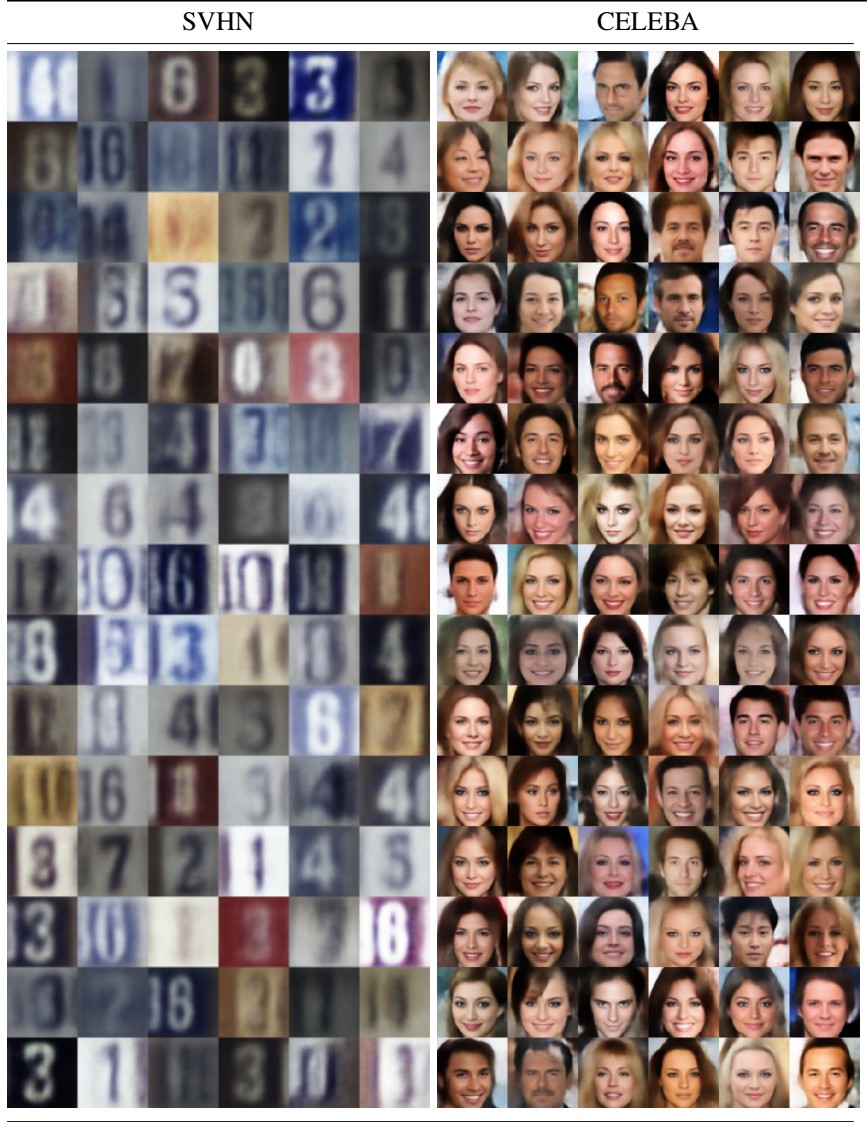

Figure 6: Additional qualitative analysis on image generation - randomly generated SVHN and CELEBA samples.