# OpenReview forum: "Shape your Space: A Gaussian Mixture Regularization Approach to Deterministic Autoencoders"
_NeurIPS.cc/2021/Conference — NeurIPS 2021 Poster_

### Official Review · Reviewer_KeyS · 2021-07-16

**Rating:** 7
**Confidence:** 4

**Summary:**

The paper builds on the ideas of Ghosh et al 2020 [10] pointing out the equivalence between Gaussian prior VAEs and deterministic AEs with noise inputation and further equivalence with regularized deterministic AEs. Since the original paper [10] needed to estimate the latent distribution ex post with a GMM, the authors here propose to use GMM prior directly and translate this into a deterministic AE with more complex regularization derived from univariate Kolmogorov-Smirnof distance combined with another term for covariance matching.They further propose a heuristic to find a reasonable heyper-parameter setting for the two regularizers. Finally, they show on a battery of experiments (generations over standard image datasets, unsupervised clustering, generating discrete structures) the favourable properties of their method.

**Limitations And Societal Impact:**

Limitations are discussed in section 5 (see above for the covariance estimation).

Societal impact - N/A

**Main Review:**

The paper proposes a logical development from Ghosh et al 2020 [10]. The move towards deterministic AEs (from stochastic VAEs) for data generation is a relatively recent direction which this paper complements. The paper is well written and easy to follow, it develops the theory and argumentation well, posing the current paper well into the existing state of the art. The idea of GMM prior is not new in the VAE setting, however, its reformulation into the regularization terms in the context of deterministic AEs is (as far as I know) novel. The motivation and the expected benefits follow in a natural way from the well explored VAE case. The empirical evaluation is thorough and on a varied set of problems. The main limitation (the need to fix the prior GMM) is important and mentioned in section 5 towards the end of the paper. It may be worthwhile mentioning it sooner and I would like to understand better how this shall be fixed in practice (how was it fixed in the experiments). Overall, I find the paper interesting, proposing an interesting and useful extension on previous research which, in my vies, can be further picked up and develop for subsequent research.

## Comments / questions
- what's the difference (if any) between your KS distance and the continuous ranked probability score (Gneiting 2004)?
- in equation (2) for the $L_{KS}$ you say you do MSE. Do you mean just squared error (SE)? I'm not sure what the mean (M) should be calculated over because your estimate of the CDF already uses all your data samples
- your $L_{CV}$ compares the sample covariance with the prior covariance - does this mean you need to estimate the sample covariance from every batch? -> effects on training stability + compute? (I didn't find this in your limitations discussion)
- is simple Frobenious norm the most appropriate for comparing the covariance matrices? (Something derived from the Wishart distribution?)

**Time Spent Reviewing:**

6

---

> ### Author Response · Authors · 2021-08-09
> **Additional clarifications**
>
> Thank you very much for reviewing our paper and for the positive remarks about our work.
>
> 1. How to fix the prior GMM - As you pointed out correctly, the necessity to fix a prior in advance to training the model is a main limitation of our approach. As suggested we will mention the importance of fixing the prior in the earlier sections in the revised version.
> In our experiments, we fixed the number of components equal to the number of classes for MNIST, FASHION-MNIST, SVHN and tuned it for CELEBA images. Nevertheless, a recent sensitivity analysis of our model towards the number of components in the GMM prior, that we added upon request of reviewer R2eb, reveals the robustness of our approach to changes in the number of components in the prior.
> Please refer to our reply to reviewer R2eb for more details.
>
>
> 2. KS distance versus continuous ranked probability score - According to the definitions, the theoretical basis of Kolmogorov-Smirnov distance and CRPS are similar. Both measure the similarity of two probability distributions, albeit in slightly different ways. The main difference lies in the applications. The KS distance is used to test
> (a) whether a set of samples is consistent with a reference distribution or
> (b) whether two sets of samples are consistent with each other, i.e., they could originate from the same distribution.
> The CRPS is used to measure the accuracy of a forecasting model for individual samples (often averaged over multiple for better accuracy).
> We will make sure to extend the discussion of the KS loss and mention other alternatives, like the CRPS or other scores (Gneiting et al. 2007).
>
>
> 3. MSE calculation for KS loss - The KS distance loss (which measures the difference in the empirical CDF and the prior's CDF) is calculated for each dimension in the latent vector and the mean value across latent dimensions is considered.
> When compared to the original KS test we employ two adaptations: (1) we replace the difference of joint CDFs by the average distance of marginal CDFs across each dimension, (2) when comparing the marginal CDFs in each dimension we replace the supremum over samples by the MSE of the differences across samples. Using the mean instead of the sum, makes the training less dependent on the batch size.
>
>
> 4. Limitations for covariance estimation - To evaluate the covariance matching loss term $\mathcal L_{CV}$, we
> calculate the empirical covariance for every batch. This might affect the training stability for higher dimensions when a relatively low batch size is considered.
> We will include this point in the discussion. Additionally, we will discuss limitations on the batch size, which can arise from the quadratic memory growth.
>
>
> 5. Frobenious norm for comparing the covariance matrices - Using the Frobenius norm in the covariance matching loss reflects our assumption that the latent dimensions should all be independent from each other and should all simultaneously match the prior's values.
> The choice for the MSE for covariance matching loss was then purely based on its prevalence in the literature. Additionally, this makes all three loss terms (reconstruction loss, KS distance loss and the covariance matching loss) behave similarly, as they are all squares.
> While we did not investigate any other metrics for matrix comparison in this scenario, exploring other options for the covariance matching is an interesting area for future studies.

---

> > ### Comment · Reviewer_KeyS · 2021-08-14
> > **Add discussion to final version**
> >
> > Thank you for your responses. Makes sense, would be good to see these reflected in the final version of the paper.

---

### Official Review · Reviewer_R2eb · 2021-07-16

**Rating:** 5
**Confidence:** 3

**Summary:**

The paper proposes a deterministic autoencoder that imposes a GMM on the latent space to learn a complex multimodal latent space. The model uses two regularizers including the KS distance and a constant covariance term to minimize the discrepancy between the marginal CDF of the empirical posterior and the given GMM prior. The authors provide multiple experimental results to show the reconstruction and sampling quality, deep clustering, and the latent model capability to generate discrete and complex structures such as chemical molecules.

**Limitations And Societal Impact:**

- The authors mentioned that some of the comparable methods suffer from computational complexity and training stability. However, the proposed model is using a sub-optimal optimization to estimate a multi-modal distribution, that may suffer from stability and convergence issues. Also, the proposed objective function requires three hyperparameters. The proposed weight estimation for the loss function seems very expensive and requires pre-trained autoencoder results.

- There is no sensitivity analysis to show the impact of the hyperparameters on the model’s performance.

- As mentioned by the authors, the proposed method greatly relies on the prior GMM, which does not exist in many domains of application. In particular, it is not clear what should be presumed for a covariance matrix.

- In Figure 1, it is not clear how each subfigure supports the outperformance of the proposed model.

- In Table 2, it is not clear what the differences between 1st, 2nd and 3rd are.

- It seems one equation (defining $\phi(\mathbf{z}_j)$) is missing in section 3.1, as mentioned in line 156.

- In Eq.9, is the term in the parentheses only a function of one $m$ or all $m=1:M$?

- In sections 4.2 and A.4, besides reporting the clustering performance, can you show the role of each dimension (as a continuous factor)? Since the covariance matrix is assumed as an identity matrix, are the latent dimensions properly independent from each other?

- I think it would be beneficial if authors compared the performance of the proposed model with a GMM prior with other mixture VAE approaches that learn mixture distributions in a disentangled way, i.e. JointVAE (Dupont, 2018), CascadeVAE (Jeong et al., 2019), especially for the clustering task.


Minor comments:
- In the related work section, there are works which are not cited in the reference section of the main text, e.g. Jakub et al.
- It seems that sometimes the bold and non-bold symbols are used interchangeably, e.g. ${F_j}^{(N)} (\mathbf{z}_{ij})$ (line 155), and ${F_j}^{(N)} (z)$ (Eq.1), or $\mu_k$ in the second part of eq.5
- In Figure 2, there is no FID scores.


**Main Review:**

Originality:
The originality of the paper seems very limited to me. Using autoencoders to fit a mixture model to the latent process to obtain the continuous and discrete representation factor has been done previously. Here, the author attempted to extend Ghosh et al., 2020 AE framework by imposing a GMM prior via minimizing the KS distance. The authors used the marginal CDF to approximate the KS distance, which is also a sub-optimal solution, without any guarantee.

Quality:
The paper does not have enough theoretical aspects. There are abundant experimental studies, but I think it would beneficial to investigate the impact of hyperparameters ($\lambda$) of the objective function and cross-correlation parameters in the prior distribution.

Clarity:
The paper is well organized. Although, I think the authors can provide a better description for figures and table captions.

Significance:
The results show the outperformance of the proposed framework for most of the datasets.


**Time Spent Reviewing:**

10

---

> ### Author Response · Authors · 2021-08-09
> **Additional ablation study on the objective function and hyperparameter sensitivity analysis**
>
> Thank you very much for reviewing our paper and for constructive remarks.
> We provide detailed answers for each of your comments below and will revise our paper submission accordingly.
>
> 1. Originality - To the best of our knowledge facilitating expressive GMM priors for deterministic autoencoders has not been attempted before. We believe that the proposed regularization loss for effectively modelling the latent space in a deterministic framework is novel.
>
>
> 2. Sub-optimal solution - As discussed in Section 3 of our paper, straight-forwardly extending the one-dimensional KS test to higher dimensions is infeasible. Therefore, we propose the efficient approximation via the marginal CDFs, which we indeed anticipate to be sub-optimal. Therefore, we propose a second regularization term to accommodate for the correlations between the latent dimensions. As discussed in Fig. 1 and the ablation study on both loss terms provided in the appendix Section A.1, this practically leads to well matched correlations across different dimensions in the prior distributions. \
> To strengthen this evaluation, we conducted an additional ablation study by training our model with only one of the both regularization terms on the full MNIST dataset.
> When training our model without the KS distance loss, we observe an FID of $49.82$. If we remove the covariance matching loss, we observe an FID of $38.45$.
> These values are significantly worse than the FID of $13.11$ that we achieve when training with the weighted combination of both regularization losses (Table 1 Samp. FID for MNIST in the main paper).
> These empirical evaluations show that the proposed combination of the two regularization terms facilitates a better prior-posterior match and hence better image generation.
>
>
> 4. Hyperparameter estimation - Regarding the loss weight estimation, we want to point out that the proposed weight estimation for the two regularization terms $\mathcal L_{KS}$ and $\mathcal L_{CV}$ in Section 3.4 of the main paper is computationally cheap and does not require pre-training an autoencoder. We propose to approximate only the weight of the reconstruction loss by training an autoencoder.
> Most importantly, this approach is much cheaper than fine tuning the regularization loss weight from scratch, because this would require multiple trainings of the full model.
>
>
> 5. Hyperparameter sensitivity analysis - From a conceptual point of view, the most important hyperparameters of our model are (a) the weights of the different terms in the training objective and (b) the number of components in the prior distribution.
> In Section 3.4 in the main paper, we propose an explicit way how to fix the weights in the loss function.
> We agree that a sensitivity analysis of the number of prior components would be beneficial.
> To investigate the sensitivity of our model to this hyperparameter, we performed a new line of experiments:
> We trained our model on the MNIST dataset using a GMM prior with $1, 5, 10, 15, 20$ and $25$ modes respectively.
> For the varying number of components we observed the following FID scores: $39.01, 21.02, 13.11, 12.84, 11.40$ and $11.98$ respectively.
> These results show that with increasing number of components in the prior the performance of our model comparatively improves (subject to a standard deviation, as e.g. $0.9$ for $10$ components, see our response to reviewer 9iT6, Q4).
> As a consequence, choosing a large number of components can only be beneficial for practical considerations.
> It would be also interesting to explore more principled approaches for choosing
> the number of components, but we leave this for future work.
> We provide all other hyperparameters to reproduce the results of the main paper in the Appendix.
>
>
> 6. GMM prior for varying applications - We believe that utilizing multi-modal prior distributions is beneficial for most applications, even for datasets with no explicit classification labels like CELEBA images.
> Even though this dataset does only comprise of face images, it is most likely not unimodal.
> Attributes like the hair, eyes etc. naturally translate into different modes.
> The main motivation to use GMM priors is to capture these complex aspects of the encoded data.
> Fixing the covariance of our prior to the identity matrix does not imply strong assumptions about the distribution of the data.
> Within each of the modes of the distribution, the covariance structure is of little practical importance, as there exists a linear transformation between the identity matrix and an arbitrary covariance matrix, which the networks can in principle learn.
>
>
> 7. Clustering performance - As mentioned in our paper, clustering is not considered as our major goal.
> We rather focus on the aspect of image and data generation.
> The mode analysis in the main paper is a mere tool to analyse the properties of the samples in each component of the GMM prior.
> While clustering approaches or methods purely focusing on latent space disentanglement (as JointVAE and CascadeVAE) aim at separating the data well within the latent space, our generative model relies on a sufficient coverage of the latent space such that clusters ideally overlap along the directions of disentanglement.
> Regarding this trade-off, a comparison w.r.t. a clustering metric is misleading.
> Within each component in the GMM prior the latent dimensions are independent from each other. We expect that each dimension in such a setting would capture one set of variations of the particular class component. For example for MNIST images, within a GMM component, one dimension corresponds to the thickness of a particular digit class while the other captures the angle of the same digit. We will include a qualitative analysis showcasing this behaviour in the revised version of the paper.
>
>
> 8. Paper clarifications -
>  * In Figure 1, we motivate the need for the second regularization term, the covariance matching loss. We would like to point out that this figure is not intended to show the out-performance of the model, but rather shows that with the combination of two regularization terms the model can overcome the limitations of the simple approximation of marginal CDFs in the KS distance loss.
> * In Table 2, we follow the same experimental setup of GrammarVAE and report the top 3 best scores observed for the corresponding optimization experiment (for arithmetic expressions the optimization objective is the $\log(1+\mathit{MSE})$ and for drug molecules this corresponds to the *water octanol partition coefficent*).
> * In Equation.9, the term in the parentheses is a function of all $m=1,\ldots,M$.
> * As mentioned in line 140,  $\Phi(z)$ corresponds to the cumulative distribution function of the normally distributed random variable $Z\sim\mathcal N(\mu, \sigma)$ in one dimension. That is, $\Phi(z)=\frac{1}{\sigma \sqrt{2\pi}}\int_{-\infty}^x\exp(\frac{-(t - \mu)^2}{2 \sigma^2})dt$.  \
> We will clarify these points in the revised version of the paper.
> We will proof read the paper again for typos and add the missing references from the related work section.

---

> > ### Comment · Reviewer_R2eb · 2021-08-25
> > **Response to the rebuttal**
> >
> > I appreciate authors' effort in addressing my concerns and running a new set of experiments. Some of the issues are clearer to me now, but some are not.
> >
> > - **“clustering is not considered as our major goal. We rather focus on the aspect of image and data generation”:** The methods that I suggested do not focus only on clustering. They also try to model the latent embedding by mixture modeling. The reason that I suggested to report the clustering performance is to evaluate the obtained latent factors in a quantitative manner, rather than a qualitative one that is reported in Figure 3. Since the main claim of the paper is using a GMM prior to efficiently shape the latent space, I think you need to provide more analyses for the latent factors, such as interpretability and disentanglement. I do not think the quality of the data generation (FID score) is enough to assess the performance of a latent representation, particularly when the comparable methods are VAE-based models which are not proposed for the high quality data generation task.
> >
> >
> > - **“Fixing the covariance of our prior to the identity matrix does not imply strong assumptions about the distribution of the data”:** In the absence of the prior (in this case the true covariance), in Eq. 3, the covariance matching loss will impose a wrong structure on the posterior. Can you show that even in the presence of a wrong prior, your model still learns the true posterior?

---

> > > ### Author Response · Authors · 2021-08-27
> > > **Additional quantitative evaluation on clustering**
> > >
> > > Thank you for the feedback.
> > >
> > > 1. As suggested, we ran a new line of experiments to provide a quantitative analysis of the clustering performance of our model.
> > > For our experiments, we evaluated the  *Unsupervised classification accuracy* - a metric that was also considered in the original CascadeVAE paper. In our initial experiments, we observed a comparable unsupervised classification accuracy for our model: MNIST - $85.53$ and FASHIONMNIST - $56.24$. For comparison, in the original CascadeVAE paper the values reported for JointVAE and CascadeVAE are as follows: MNIST - $78.33, 84.19$ and FASHIONMNIST - $51.51, 57.72$ respectively. We believe that the distance between the modes is a deciding factor for better clustering performance.
> > > We will provide an ablation study on the distance between the modes, clustering performance and the image generation performance in the revised Appendix.
> > >
> > > 2. Thank you very much for elaborating on this question; we might not have addressed this clearly enough in our first answer. For CELEBA, as we don't have a more informed prior, we resolve to using the identity because it is the simplest option. We observe that it works well in practice. Yet, we agree that, in general, one is never sure whether an imposed prior is correct. We will discuss this aspect in more detail in the limitation section.

---

> > > > ### Author Response · Authors · 2021-09-01
> > > > **Discussion period about to end**
> > > >
> > > > Since the discussion period is approaching its end in less than 48 hours, we want to consult whether you have had a chance to look at our new experimental results and whether there are any further concerns that we could address?

---

> > > > > ### Comment · Reviewer_R2eb · 2021-09-01
> > > > > **Response to the follow up**
> > > > >
> > > > > I have read the latest responses. Overall, I think the authors did put a lot of effort into this work and I have increased my score, but I cannot give higher. I still think the paper's originality and contribution are limited.

---

### Official Review · Reviewer_9iT6 · 2021-07-16

**Rating:** 6
**Confidence:** 4

**Summary:**

The author(s) motivate and derive a novel deterministic auto-encoder loss that naturally encourages a multi-model latent representation that follows a GMM. The author(s) motivate and derive this loss using the KS test metric and generalize it to multiple dimensions and multiple modes. Their experiments, whereby they ancestrally resample latent codes show an impressive ability to generate realistic samples.

**Limitations And Societal Impact:**

The author(s) sufficiently address limitations. Although, I do wonder what would happen under class imbalance. For instance say the data naturally falls into two clusters, a majority and minority. Do the author(s) have reason to believe their methods would perform well in this scenario? I think this question could be important to understand potential negative societal impacts.

**Main Review:**

### Originality:
- To the best of my knowledge, the proposed regularization terms (equations 2, 3, 6, 7) are novel.

### Quality:
- The new regularization terms are well motivated using the KS test.

### Clarity:
- Paper is generally well written.
- Ancestral resampling from prior does not necessarily generate new samples from the training distribution (lines 25-26) if the latent space utilization is multi-modal.
- VAE paragraph references are inconsistent. Sometimes [x] is used other times NAME, et al. is used. Please be consistent. Anytime a name is used there is also no link to the appendix making reference checking cumbersome.
- Equation 9 sets $\lambda_{KS}^{-1}$ twice. I imagine the second one is supposed to be $\lambda_{CV}^{-1}$
- My major concern is how sampling FID (Samp.) is performed across the methods. Line 219 has "models/by" does the '/' denote 'or' or is suppose to be space? If all methods fit Gaussian to latent codes and sample from that, then this seems fair. But if some methods use GMM priors for sampling whereas others use uni-modal distributions, then this comparison seems unfair.
- For the VAE methods, do the author(s) sample the decoder distribution or report its mean/mode?

### Significance:
- Table 1 results are strong, but lack error bars, which is moderately concerning. In particular, I wonder if some of the GMM wins are statistically insignificant.

### Other:
I love that "Since the cluster is part of a carbon-neutral framework, these experiments did not contribute to climate change."

**Time Spent Reviewing:**

2

---

> ### Author Response · Authors · 2021-08-09
> **Error bars for Table 1**
>
> Thank you very much for your comments and the positive review of our paper, which rates our regularization approach for deterministic autencoders as novel and well motivated.
>
> 1. Ancestral re-sampling from prior if latent space utilization is multi-modal  - We are referring to the uni-modality assumption in the lines 25-26 in the main paper. Unfortunately, we are not sure if we understand your first question regarding the "ancestral resampling" from the latent prior correctly. Could you please elaborate on this question and/or provide us a reference that can help us clarify this?
>
> We gladly address your remaining question below.
>
> 2. Sampling FID method - To ensure a fair comparison of all methods, we calculate the sampling FID in Table 1 by fitting a unimodal Gaussian to the latent codes for all models. That is a very important point, which we have not emphasized clearly enough.
> We will clarify this in the revised version of our paper.
>
>
> 3. We report the FIDs for VAE methods by sampling from the prior distribution (isotropic Gaussian for VAE/WAE, another VAE for 2SVAE).
>
>
> 4. Error bars for Table 1 - The FIDs observed by sampling from the prior along with error bars (for $5$ different runs) are as follows, MNIST: $13.11\pm0.9$, FASHION-MNIST: $33.70\pm0.8$, SVHN: $37.42\pm1.1$ and CELEBA: $49.79\pm1.2$. And the FIDs that we observe after fitting a GMM to the latent space of our model are as follows (for $10$ different runs), MNIST: $12.82\pm0.6$, FASHION-MNIST: $26.62\pm0.8$, SVHN: $36.46\pm0.9$ and CELEBA: $44.79\pm1.0$.
> We will add these results to the revised version of our paper.
>
>
> 5. Class imbalance in the data - Thank you for raising the question about class imbalance in the data. This is indeed a failure case which we have not addressed, but which could lead to unforeseen consequences. We would expect the model to separate the two classes if the imbalance is weak and the classes are sufficiently different such that the reconstruction loss outweighs the regularization penalty for the mismatch. Extending our prior to accommodate for this by introducing a weighted GMM prior is a very interesting direction for future work.
>
> Thank you for pointing out the inconsistency in our references. We will correct this and the typos in the revised version.

---

> > ### Author Response · Authors · 2021-09-01
> > **Discussion period approaching its end**
> >
> > Since the discussion period is approaching its end in less than 48 hours, we humbly consult whether you have had a chance to look at our answer. Are there any further concerns that we could address?

---

### Official Review · Reviewer_vNcb · 2021-07-18

**Rating:** 6
**Confidence:** 3

**Summary:**

This paper proposes to learn GMM priors in autoencoders, to be able to construct a generative model. They compare the generation quality of the proposed approach with VAEs using the FID metric. One interesting contribution seems to be the fact that they are using a loss function other than the GMM likelihood to fit the GMM.

**Limitations And Societal Impact:**

The authors have a small paragraph to discuss the limitations of their work. I am not entirely convinced that it is a comprehensive discussion on the limitations.

They do not discuss the societal impact, but I do not think there is an outstanding aspect to this work that makes it relevant for such discussion.

**Main Review:**

The proposed approach is definitely not entirely novel. I might be missing some aspects of it that is novel, but in general using GMMs or its variants has been used in the literature.

The results with the proposed approach seems to be better than few alternatives in terms of FID scores, but I would like to note that the datasets on which the authors try their approach is are relatively old. I think it would be worth exploring more challenging datasets which larger images such as celeba-hq, or larger images such as Imagenet like it is used in this paper https://arxiv.org/pdf/1906.00446v1.pdf
would be required to further convince the audience of the merit of the approach.

The paper also contains number of typos, and these should be corrected in an eventual acceptance in a conference.

**Time Spent Reviewing:**

2

---

> ### Author Response · Authors · 2021-08-09
> **Novelty, limitations and additonal remarks on empricial evaluation**
>
> Thank you very much for reviewing our paper and for your comments.
>
> 1. Novelty - As pointed out correctly, our main contribution is the proposed regularization loss for deterministic autoencoders, that we derive from the Kolmogorov-Smirnov test for equality of probability distributions.
> As mentioned in the related work of our paper, expressive priors like GMMs have been explored before in the VAE setting.
> To the best of our knowledge, translating this idea into a regularization loss for deterministic autoencoders is novel.
> In our paper, we employ the proposed novel regularization loss to train a deterministic auto-encoder to shape the latent space effectively.
> We consider the proposed method as an extension to the recent research in this field. Compared to the recent work on determninistic autoencoders (Ghosh et al 2020 [10]), we do not require an additional step to generate new samples.
> We also propose an effective method to fix the hyperparameters of the proposed regularization loss terms.
> The empirical evaluations show the potential of the method to effectively structure the image space as well as complex discrete spaces when compared to the baseline approaches.
>
>
> 2. Additional datasets  - Considering more complex datasets would be obviously interesting for image generation applications. At the same time, we would like to point out that in the paper we focus on showing the generalization ability of the model to effectively shape the latent space of continuous as well as complex discrete domains.
> Hence, for image generation we chose four different types of standard image datasets, and for modeling discrete domains we consider two optimization problems, (a) arithmetic expression fitting and (b) chemical drug molecule design, which is of great interest to the research community. \
> Thank you for pointing out the VQ-VAE (van den Oord et al., 2017; Razavi et al., 2019) reference, we will include a discussion of this work to the related work section of the paper.
> The VQ-VAE model can be also considered as a deterministic autoencoder and it focuses on high fidelity image generation. Similar to RAEs (Ghosh et al 2020 [10]), training VQ-VAE involves two stages of training relying on complex discrete
> autoregressive density estimators.
> Moreover, the training loss of VQ-VAEs is non-differentiable due to the quantization of the latent vector.
> Hence, these models are not suitable for the optimization experiments considered in our paper.
>
>
> 3. Limitations not comprehensive - In the main paper we focus on a discussion of conceptual limitations of our approach.
> We discuss the necessity to fix a prior in advance before training the model. Further, we point out conceptual limitations that might raise from simplifying the original distance metric used in the Kolmogorov-Smirnov test.
> As mentioned in the other reviews, a further point of discussion would be how class imbalance in the training data would effect the performance of our approach and the limitations for covariance estimation.
> We discuss this point in the replies to reviewers 9iT6 and KeyS and will add it to the main paper accordingly. \
> Please refer to the replies to both reviews for a detailed discussion of all points.

---

> > ### Comment · Reviewer_vNcb · 2021-08-18
> > **Comparison with GMM-prior VAEs?**
> >
> > Thanks for your reply.
> >
> > I am thinking along the lines of the following:
> > 1) If I am not missing it, it seems you haven't qualitatively or quantitatively compared with an autoencoder where the GMM prior is trained with a regular GMM likelihood. For instance in table 1, it would be good to know how the numbers compare with your proposed regularization loss and the standard approach.
> > 2) You are saying that VQ-VAE is not suitable for your experiments for comparison, but it can for instance be trained, so that you can include it on table 1 for example, right?

---

> > > ### Author Response · Authors · 2021-08-23
> > > **Additional baseline comparison results for Table 1**
> > >
> > > Thank you for the feedback.
> > >
> > > 1.  Autoencoder where the GMM prior is trained with a regular GMM likelihood - Thank you for suggesting this baseline experiment for image generation. As suggested, we trained a variational autoencoder with a regular GMM likelihood and observed the following performance. For MNIST, FASHIONMNIST, SVHN and CELEBA - the sampling FIDs are $21.35, 40.23, 49.74$ and $65.35$, reconstruction FIDs are $20.64, 38.79, 48.65$ and $64.22$, interpolation FIDs are $20.21, 38.54, 47.15$ and $64.92$ respectively.
> > > We will update Table 1 accordingly in the revised version of the paper.
> > >
> > > 2. VQ-VAE for image generation experiments - Unfortunately, the original VQ-VAE paper does not provide experimental results on widely used low resolution datasets such as MNIST, SVHN, CELEBA etc. In an initial experiment, we applied VQ-VAE to MNIST data. However, the originally used bottleneck in VQ-VAE is overly large for such low resolution data. We therefore ran initial experiments with different hyperparameter settings for the size of the discrete latent space/number of embeddings $K$ and the dimensionality of each latent embedding vector $D$. \
> > > Our first results are as follows:\
> > > for $K=5$ and $D=64$, we observed a sampling FID  of $49.76$, reconstruction FID of $48.27$ and interpolation FID of $47.19$; \
> > > for $K=10$ and $D=64$, we observed a sampling FID of $33.21$, reconstruction FID of $31.26$ and interpolation FID of $32.45$; \
> > > for $K=32$ and $D=128$, we observed a sampling FID of $18.51$, reconstruction FID of $17.62$ and interpolation FID of $16.85$. \
> > > These results show that VQ-VAE hyperparameters need further tuning, which we are currently doing. At the same time, we are computing the more expensive original setting with number of embeddings, $K=512$, which might however cause overfitting on MNIST. We will report the best results we can achieve in the final version.\
> > > We would also like to clarify why, in our original understanding, the direct comparison in the context of image generation did not make sense, as we stated in our first response. We understand that VQ-VAE proposes to quantize, i.e. discretize, the latent space using regularization terms, such as to allow for high quality image generation. To this end, it is also using as a backbone network a Resnet-architecture. However, VQ-VAE does not address the question of how to structure the continuous latent space before quantization. In contrast, the proposed approach addresses specifically this question of how to efficiently structure the latent space. In that sense, the two approaches are rather complementary than direct competitors. A combination of both for high quality image generation from pre-structured latent spaces prior to quantization would be an interesting topic for future research.
> > >
> > > --------------------------------------------------------------------------------------------------------------------------------------------------------------------------------------
> > > Update: For the original hyperparameter configuration in VQ-VAE paper, $K=512$ we observed the following performance on MNIST images: sampling FID  of $20.43$, reconstruction FID of $19.05$ and interpolation FID of $18.99$.
> > > Better results might probably require more tuning.

---

> > > > ### Author Response · Authors · 2021-09-01
> > > > **Following up on your questions**
> > > >
> > > > Since the discussion period is approaching its end in 48 hours, we humbly consult whether you have had a chance to look at our answer including the results of the additional baseline comparison; and if that could make you willing to raise our score. We are also happy to address any further concerns.

---

> > > > > ### Comment · Reviewer_vNcb · 2021-09-01
> > > > > **Thanks for your response**
> > > > >
> > > > > I appreciate the efforts of the authors, and consequently raising my assessment by one point.

---

### Decision · Program_Chairs · 2021-09-27

**Decision:**

Accept (Poster)

**Comment:**

By the scores alone, this paper is just borderline. Reviewers had a number of questions and clarifications, which were responded to well during the author response period, and even included additional results against a VQ-VAE baseline; this led to multiple reviewers raising their scores. Leaving aside issues of clarity (which could be fixed by camera-ready, given the discussion here), the main concern across the reviewers is the degree of novelty and originality, given that on the one hand there are a number of other methods which use Gaussian mixture priors on VAEs, and on the other hand this is also a direct extension of Ghosh et al 2020. The other concern is the somewhat heuristic use of an approximation to the KS statistic in the loss, and a question of how sub-optimal the result is, as well as whether using simply a Gaussian mixture likelihood could not lead to similar results.